# Path Planning and Bending Behaviors of 3D Printed Continuous Carbon Fiber Reinforced Polymer Honeycomb Structures

**DOI:** 10.3390/polym15234485

**Published:** 2023-11-22

**Authors:** Kui Wang, Depeng Wang, Yisen Liu, Huijing Gao, Chengxing Yang, Yong Peng

**Affiliations:** Key Laboratory of Traffic Safety on Track of Ministry of Education, School of Traffic & Transportation Engineering, Central South University, Changsha 410075, China; kui.wang@csu.edu.cn (K.W.); wangdepeng@csu.edu.cn (D.W.); ghjghj@csu.edu.cn (H.G.); yong_peng@csu.edu.cn (Y.P.)

**Keywords:** 3D printing, continuous fiber reinforced polymer, honeycomb structures, path planning, structural defects, bending behaviors

## Abstract

Continuous fiber reinforced polymer composites are widely used in load-bearing components and energy absorbers owing to their high specific strength and high specific modulus. The path planning of continuous fiber is closely related to its structural defects and mechanical properties. In this work, continuous fiber reinforced polymer honeycomb structures (CFRPHSs) with different printing paths were designed and fabricated via the fused deposition modeling (FDM) technique. The investigation of fiber dislocation at path corners was utilized to analyze the structural defects of nodes caused by printing paths. The lower stiffness nodes filled with pure polymer due to fiber dislocation result in uneven stiffness distribution. The bending performance and deformation modes of CFRPHSs with different printing paths and corresponding pure polymer honeycomb structures were investigated by three-point bending tests. The results showed that the enhancement effect of continuous fibers on the bending performance of honeycomb structures was significantly affected by the printing paths. The CFRPHSs with a staggered trapezoidal path exhibited the highest specific load capacity (68.33 ± 2.25 N/g) and flexural stiffness (627.70 ± 38.78 N/mm). In addition, the fiber distributions and structural defects caused by the printing paths determine the stiffness distribution of the loading region, thereby affecting the stress distribution and failure modes of CFRPHSs.

## 1. Introduction

Due to their lightweight, high strength, and superior energy absorption capacity, honeycomb structures are widely used as load-bearing components and energy absorbers in the forms of sandwich structures and thin-walled structures, owing great application potential in many industrial fields such as aerospace, automobile, and construction [1,2,3,4,5]. The macroscopic performance of honeycomb structures depends on the structural design of their cells and material selection [6,7,8]. With the advancement of design, numerous cells with various forms have been applied in the evolution of honeycomb structures, such as Kagome [9], re-entrant [10], chiral honeycombs [11], etc. The materials of honeycomb structures have also expanded from traditional metals and wood to various newly developed composites, such as polymer composites with excellent mechanical properties and lightweight performance [12,13,14].

Adding continuous fibers as reinforcement in polymers could significantly improve the specific strength and modulus while maintaining their lightweight performance [15,16,17,18]. Traditional preparation methods of continuous fiber reinforced polymers (CFRPs), such as hot press molding [19], vacuum-assisted molding [20], and filament winding [21], require complicated multi-step processes and expensive manufacturing equipment, while the manufacturing of honeycomb structures commonly needs custom molds and subsequent bonding processes [22,23]. These factors became obstacles to the development of continuous fiber reinforced polymer honeycomb structures (CFRPHSs). However, 3D printing techniques with high design freedoms and multi-scale molding capabilities allow a non-mold-based approach to manufacturing structures with customized materials and complex geometric shapes, providing the possibility for the integrated manufacturing of CFRPHSs [24,25,26]. According to the ISO/ASTM 52900 standard, 3D printing technologies are divided into seven types, among which material extrusion (MEX) technology is very suitable for 3D printing of polymers [27,28]. Fused deposition modeling (FDM), as one of the most widely used MEX technologies, with rapid prototyping and low-cost characteristics, combines continuous fibers with polymer matrix [29,30,31]. Many researchers have introduced CFRPs with lightweight and high specific strength features into the design of honeycomb structures via the FDM technique [32,33,34]. For example, Kentaro et al. [35] printed a series of CFRPHSs with hexagon, rhombus, rectangle, and circle core shapes through the FDM technique. The experimental results showed that bending properties increased with the increase of core density, and the CFRPHSs with a rhombus core shape exhibited the strongest bending properties. Dou et al. [36] fabricated hexagon-filled CFRPHSs and studied their in-plane compression properties. The results showed the specific energy absorption of composite honeycomb was 186.58% and 596.84% higher than that of pure polymer matrix and aluminum alloy, respectively.

In addition, the printing path also had a crucial influence on the arrangement and orientation of continuous fibers and further affects the mechanical properties of the structure [37,38,39]. Some researchers have tried to improve the manufacturing process and mechanical properties of CFRPHSs by printing path planning [40,41]. For most FDM-based 3D printers without fiber cutting and jumping functions, the one-stroke printing path planning was an inevitable choice to achieve cyclic printing in the height direction [42,43]. Based on one-stroke printing path planning, Quan et al. [44] designed and fabricated a group of auxetic honeycomb-filled CFRPHSs. The compressive stiffness and energy absorption of the CFRPHSs were increased by 86.3% and 100%, respectively, compared to the structures made of pure polymer matrix. Dong et al. [45] investigated the influences of printing paths on the tensile properties of diamond-filled CFRPHSs. The improved trapezoid-like path allowed an even fiber distribution without weak nodes in the CFRPHSs, exhibiting excellent tensile strength. 

Depending on the path planning, there were various situations of the fiber crossing at the nodes of the honeycomb structures [46]. The nodes without fiber interlacement might result in structurally weak regions [45], and printing path corners inevitably appeared at the nodes owing to the structural features of the honeycomb. Some printing defects, including fiber dislocation and the absence of fiber, can occur at the corner paths, which might further exacerbate the structural defects of the CFRPHSs [47,48,49,50]. The details of the fiber distributions and printing defects were effectively displayed on the honeycomb structures, especially at the nodes. These fiber distributions and structural defects caused by printing paths affect the mechanical behavior of CFRPHSs during loading. However, to the best of the authors’ knowledge, only a small amount of literature has investigated how the printing path planning affects 3D printed continuous fiber reinforced structural defects and further mechanical properties.

In this study, diamond-filled CFRPHSs with four different printing paths were designed and fabricated. Three corners (45°, 90°, and 135°) that appeared in the printing paths were selected to investigate the fiber dislocation phenomena at the path corners. Three-point bending tests and morphological analysis were conducted to investigate the influence of the printing path and printing defects at path corners on the bending behaviors of CFRPHSs. This study analyzed the mapping relationship between the printing path planning, structural defects, and final structural and mechanical performance, aiming to provide guidance for the printing path planning of 3D printed CFRPHSs with high performance.

## 2. Materials and Methods

### 2.1. Design and Fabrication

In this study, a honeycomb structure with a diamond configuration was designed to investigate bending behaviors and failure mechanisms, as shown in Figure 1a. The honeycomb core consisted of a repeating arrangement of unit diamond cells. The included angle of the diamond cell was 90°; the length (*L*) and width (*W*) of the CFRPHSs were determined by the diamond cell length (*P*); the height (*H*) of the CFRPHSs was 10 mm.

Polyamide (PA) filament (Polymaker Inc., Changshu, China) with a diameter of 1.75 mm was adopted as the polymer matrix of the CFRPHSs. The continuous carbon fiber (CCF) (HTA40-E15-1K, Toho Tenax Co., Ltd., Tokyo, Japan) was used as the reinforced fiber, which was a flat fiber bundle with a width of 0.80 mm. Before printing, the filaments were sealed and stored at 20 °C. According to the instructions from the materials supplier, the mechanical properties of the PA filament and CCF are listed in Table 1. 

A commercial 3D printer (COMBOT-200, Fibertech Technology Development Co., Ltd., Shaanxi China), based on the FDM technique, was used to fabricate the CFRPHSs. The schematic representation of the 3D printing process is shown in Figure 1b. The PA filament (blue) and CCF (black) were simultaneously fed into the extrusion head through these two feed ports, respectively. The in situ impregnation of CFRPs was realized in the heated mixing chamber. Then, CFRPs were extruded onto the printing bed and solidified rapidly. With the relative motion of the printing nozzle and the printing bed, the subsequent CFRPs were dragged out from the printing nozzle, ensuring the continuity of the printing process. The models of the CFRPHSs were sliced before manufacturing and the slice data was recorded in the G-codes developed by our team, including the printing paths, geometrical parameters, and printing parameters. For all samples in this study, the printing nozzle temperature was 260 °C, and the printing bed temperature was 50 °C. The printing speed was 180 mm/min, and the layer height was 0.4 mm.

One-stroke printing path planning was required for the CFRPHSs to ensure the integrity of CCF due to the printer’s lack of jumping and fiber-cutting features [51]. The manufacturing of 3D printed CFRPHSs was based on the layer-by-layer stacking mechanism of FDM, which was only needed to design a single-layer printing path and print the same single-layer printing path repeatedly in Z-direction. Based on the flexibility of the 3D printing path planning, the printing paths at the nodes lapped between the core and face sheets of the CFRPHSs could be crossing or non-crossing, resulting in fiber interleaving or adjacency [45,46]. Figure 2a shows three printing path strategies at the nodes. According to these different path strategies, four printing paths for the diamond-filled CFRPHSs were designed in this study, as shown in Figure 2b–e. These complete single-layer paths were decomposed into multiple branches distinguished by different colors according to the printing order; these branches were printed successively from the outside to the inside. The combined graphs of the path branches were shown at each bottom. The printing nozzle began from the start point of a single-layer printing path and moved along the arrow direction during printing. After completing the one-stroke path of each layer, the printing nozzle returned to the start point. Then, the printing bed moved downward by a layer height along the Z-direction to repeat the print for the next layer. In the whole printing process, the continuous fibers remained intact. On the right side of these printing paths, the corresponding fiber distribution modes at the nodes were displayed, including antagonistic distribution, symmetrical distribution, and cross-distribution. In addition, pure polymer honeycomb structures (PPHSs) were also prepared with the same geometrical parameters and printing parameters.

### 2.2. Experimental Methods

The fiber dislocation inevitably appeared at the path corners in the CFRP’s printing process. The morphology of the fiber dislocation at different path corners was recorded by a digital camera (5D mark IV, Canon, Tokyo, Japan). ImageJ version 1.53t software was used to measure the actual angle and position of the fibers at these corners, and a quantitative evaluation method of fiber dislocation was used to analyze the obtained data.

Honeycomb structures are subjected to a complex stress state during the bending process, and these are helpful in studying the influence of the printing path on mechanical behaviors [52,53]. Therefore, the three-point bending tests were performed on a universal material testing machine (E44, MTS Co., Eden Prairie, MN, USA) to investigate the bending properties and failure modes of 3D printed CFRPHSs with different printing paths and PPHSs. The schematic diagram of the three-point bending test is shown in Figure 3. The span between the supports was 80 mm, and the loading roller radius was 5 mm [35,54]. The displacement loading was applied to the samples with a speed of 2 mm/min, and the maximum displacement was 20 mm. The load and displacement data of all tests were automatically collected in the data acquisition system, and the whole test process was recorded by the digital camera. The failure areas of the samples were characterized through an optical microscope (AO-3M150GS, AOSVI, Shenzhen, China). According to the measured load-displacement curves, several indicators were used to quantify the bending properties, including specific load capability and flexural stiffness. The specific load capability (SLC) was calculated by dividing a maximum peak load by the structure mass:(1)$$SLC=\frac{F_{peak}}{m}$$
where m is the mass of the sample, and $F_{peak}$ is the maximum peak load. The flexural stiffness was calculated as the slope of the load-displacement curve in a low displacement regime where the curve was almost linear. Each group of tests was repeated five times to ensure reliability, and the average results are reported in this study.

## 3. Results and Discussions

### 3.1. Investigation of Fiber Dislocation at Path Corners

The corner paths appear at the nodes (i.e., the corners of the honeycomb walls) due to the structural features of honeycomb structures. At the nodes lapped between the core and face sheets in the CFRPHSs, the printing paths are composed of different corner paths distributed in different forms (e.g., crossing or non-crossing), which are the distinctions between different printing paths of the CFRPHSs [45,46]. The printing defects of fiber dislocations are inevitably generated at the corner paths [48,49]. The corner paths with fiber dislocation may cause structural defects at the nodes of the CFRPHSs, which can further affect the mechanical properties and deformation modes. 

The degree of fiber dislocation was mainly affected by the angle of path corner and the shortest length of straight paths forming the corner [47]. For the diamond-filled CFRPHSs, there were only three kinds of path corners (45°, 90°, and 135°) in all printing paths, as shown in Figure 4a. The shortest straight path ($L_{s}$) of the CFRPHSs was limited by the structural geometric parameters. Here, $L_{s}$ was equal to the diamond cell length (*P*), as well as the width of the CFRPHSs (*W*). Thus, a corner path consisting of two equal straight paths ($L_{s}$) is designed in Figure 4b, which could be 45°, 90°, and 135°. Under the default printing process parameters, these corner paths were printed as $L_{s}$ = 10 mm, 15 mm, and 20 mm, corresponding to Figure 4d–f, respectively. Noticeable carbon fiber dislocations at these path corners could be observed. To further quantify the degree of fiber dislocation at these corners and determine the appropriate geometric parameters for the CFRPHSs, the angle distortion ratio (φ) and the height distortion ratio (δ) were used as the quantitative evidence, and are expressed as follows:(2)$$\varphi =\frac{\alpha_{1}-\alpha_{0}}{\alpha_{0}}$$
(3)$$\delta =\frac{\varepsilon }{h}$$
where $\alpha_{0}$ denotes the programmed angle of the path corner, $\alpha_{1}$ refers to the actual angle of the CCF at the path corner, h is the theoretical height of the CCF raised at the corner, and ε is the retraction distance of the CCF at the corner, as shown in Figure 4c.

According to Formulas (2) and (3), the calculated values of φ and δ of these path corners are exhibited in Figure 5a,b, respectively. It can be observed that the values of φ and δ increased with the decrease of $\alpha_{0}$, as well as the decrease of $L_{s}$. According to the schematic diagram in Figure 6a, the incompletely solidified matrix could not constrain the fiber effectively [47], which was the main factor causing the fiber dislocation at the corner paths. A pulling force F, generated by the movement of the nozzle during printing, pulled the subsequent CCF out of the nozzle to ensure the continuity of the printing. After the nozzle turned, the direction of the pulling force F changed, which was parallel to the second straight path. However, there was some incompletely solidified matrix close to the printing nozzle due to the short cooling time and inadequate solidification, which could not provide enough constraints for the CCF at the corner. As a result, the printed CCF at the end of the first straight path was pulled out by the pulling force F. The direction of the first straight path and its vertical direction were defined as the X-direction and the Y-direction, respectively. F_x_ and F_y_ are the two vertical components of the pulling force F in the X-direction and Y-direction. When $\alpha_{0}$ ≥ 90°, the fiber dislocation is mainly caused by F_y_, while F_x_ along the X-axis positive direction has no negative effect. The vertical component F_y_ increased as $\alpha_{0}$ decreased when F was constant, exacerbating the fiber dislocation. When $\alpha_{0}$ < 90°, the CCF, pulled by F_x_ along the X-axis positive direction, and F_y_, exhibited significantly higher angle distortion and height distortion. Therefore, the values of φ and δ were increased with the decrease of $\alpha_{0}$, and they were significantly higher when $\alpha_{0}$ = 45°. The length of the incompletely solidified matrix close to the nozzle was almost fixed under the same process conditions. The proportion of the fiber dislocation area in the corner increased with the decrease of $L_{s}$ when the angle was constant. Therefore, both the values of φ and δ increased with the decrease of $L_{s}$, as shown in Figure 5. 

In addition, the fiber could not maintain a fixed position relative to the nozzle port, which was another factor causing fiber dislocation [48], as shown in Figure 6b. To reduce fiber damage during printing, there was a difference between the cross-sectional area of fiber and the nozzle port, resulting in the CCF with a high degree of freedom in the nozzle port. The relative position of CCF could not remain fixed during nozzle turning owing to the high fluidity of the molten polymer matrix. Therefore, the CCF could not reach the top of the corners. 

According to the above investigation, the angle distortion ratio of the 45° corner reached 133.1%, and its height distortion ratio reached 45.7% when $L_{s}$ = 10 mm, showing serious printing defects. To maintain the fiber dislocation at relatively low levels, $L_{s}$ could not be less than 15 mm. Moreover, the CFRPHSs show better mechanical properties with increasing filling density, which tended to be with a smaller $L_{s}$ [55]. Therefore, $L_{s}$ was set as 15 mm of the CFRPHSs with different printing paths.

### 3.2. Effect of Printing Path on Bending Properties

The distribution and orientation of continuous fibers within the CFRPHSs were determined by their printing paths, which affects the mechanical behaviors [56]. To investigate the influence of the printing path on the bending performance of the 3D printed CFRPHSs, three-point bending tests for the CFRPHSs with four printing paths and the PPHSs were carried out. According to the discussions in Section 3.1, *L* and *W* of the CFRPHSs were set as 120 mm and 15 mm, respectively. The maximum bending displacement was 20 mm. Figure 7a,b shows the photographs of the bending test samples at displacements of 0 mm and 20 mm, respectively.

Figure 8a exhibits the load-displacement curves of the CFRPHSs with four printing paths and the PPHSs under three-point bending tests. It can be observed that the load-displacement curves of these CFRPHSs can be divided into three clear stages: the elastic stage, yield stage, and plateau stage. The CFRPHSs initially exhibited short-term elastic deformation during the bending process, and the load increased almost linearly up to the initial peak load, which was the elastic stage. After elastic deformation, the CFRPHSs quickly entered the yield stage. In this stage, the load dropped rapidly to a relatively low value with the progressive failure of structures. Subsequently, the load-displacement curves presented a plateau stage accompanied by small fluctuations in the load. Meanwhile, the deformation stages of the PPHSs were similar to those of the CFRPHSs.

The specific load capability and flexural stiffness of these CFRPHSs and the PPHSs were calculated to further evaluate the bending properties, as shown by the results in Figure 8b and Table 2. It was observed that the addition of continuous fiber greatly improves the flexural stiffness of honeycomb structures. In particular, the CFRPHSs with path C exhibited the highest flexural stiffness (627.70 ± 38.78 N/mm), which was 241.8% higher than the PPHSs. However, the enhancement effect of continuous fiber on specific load capability was more significantly influenced by the printing paths. The CFRPHSs with path C showed the highest specific load capability (68.33 ± 2.25 N/g), while the specific load capability of the CFRPHSs with path B (44.12 ± 3.04 N/g) was close to that of the PPHSs (43.48 ± 2.46 N/g). Therefore, the printing paths would affect the improvement effect of continuous fibers on the bending performance of honeycomb structures. 

During the bending process, the upper face sheet was directly subjected to the load, whereas the honeycomb core showed supporting and strengthening effects for the face sheet to slow the deformation [57]. Figure 9 further exhibits the fiber distributions of the CFRPHSs in the loading region, distinguished by the color lines. There were three fiber distribution modes at the red node beneath the loading roller, including antagonistic distribution, symmetrical distribution, and cross-distribution, as shown in Figure 9a. The symmetrical distribution consisted of two opposite and symmetrical 45° path corners corresponding to path B. Based on the discussions in Section 3.1, the fiber dislocation of the 45° path corner was the most severe. Thus, a large area of polymer matrix without fiber filling formed at the red node of the CFRPHSs with path B, resulting in a lack of direct support for the compression area of the upper panel because of the low stiffness of pure polymer. It caused the lowest flexural stiffness and specific load capacity of path B. The antagonistic distribution was a 90° path corner with the tip facing the upper face sheet, corresponding to path A. And the cross-distribution was composed of two intersecting 135° path corners, corresponding to paths C and D. The red nodes with these two fiber distribution modes had higher stiffness than the nodes filled with pure polymer and could provide better support to the loading region of the upper face sheets, causing higher flexural stiffness and specific load capacity of the CFRPHSs compared with path B. 

In addition, the load acting on the upper face sheet was transmitted to the lower face sheet through continuous fibers distributed in the core. The fiber distribution modes (45°, 90°, and 135° corners) close to the lower face sheet would affect the load transmission from the core to the lower face sheet, as shown by the green nodes in Figure 9b. Among them, the CCF distributed at 135° corner had the largest contact area with the lower face sheet in path C, which could better transfer the load to the lower face sheet and obtain the support from the lower face sheet. Moreover, the staggered trapezoidal path C allowed CCF intersections at all nodes of the CFRPHSs, ensuring a strong connection between core and face sheets [58]. These enabled the structure to have good load transfer performance and fully utilize the supporting function of the honeycomb core. Thus, the CFRPHSs with path C showed the strongest specific load capacity and flexural stiffness compared to other CFRPHSs and PPHSs.

The core and face sheets of honeycomb structures were subjected to a complex stress state during the bending process, accompanied by potential failure modes, including face sheet indentation and core shear. Figure 10 shows the bending deformation states of the PPHSs at displacements of 6 mm and 18 mm, corresponding to the yield stage and the plateau stage, respectively. In Figure 10a, plastic yielding occurred in the lower face sheet near the node, causing the load to begin to decrease. When the displacement reached 12 mm, the PPHSs exhibited indentation failure and core shear failure induced by the core upper member buckling, as shown in Figure 10b. The plastic yielding of the lower face sheet was deepening, accompanied by fracture and delamination. Due to the contact between adjacent honeycomb core members and face sheets, the rate of load decline slowed down, and the PPHSs entered the plateau stage.

Figure 11 and Figure 12 show the photographs of the bending deformation states of these CFRPHSs at two different moments during the yield and plateau stages, which corresponded to the colored lines in Figure 8a, to further analyze their failure modes. The CFRPHSs at the state I began to enter the yield stage. It can be observed in Figure 11 that the upper face sheet bending beneath the loading roller occurred in all CFRPHSs. Meanwhile, the core shear failure began to occur in the CFRPHSs, resulting in a decrease in their structural bearing capacity. Two possible core shear failure modes are caused by core member buckling or core member yielding, respectively [54]. For paths A, C, and D, the initial failure modes were core shear failure caused by core member buckling, as shown in Figure 11a,c,d. The degrees of core member buckling in these CFRPHSs were different. Path C, with the most severe core member buckling, resulting in the most rapid drop in state I in the load-displacement curve. However, for path B, two 45° corners with severe fiber dislocation were distributed close to the lower face sheet (see Figure 9b), where the nodes filled with pure polymer had low stiffness, resulting in yield failure easily occurring at the end of core members, as shown in Figure 11b. The core shear failure caused by yielding had a smaller effect on the support of the honeycomb core to the upper face sheet than that caused by buckling, maintaining the load of path B declining at a slower rate. Therefore, the core shear failure modes of the CFRPHSs were affected by fiber distribution modes at the nodes. The rate of load decline in the yield stage was determined by the mode and degree of core shear failure.

At state II, the CFRPHSs began to enter the plateau stage. Indentation failure occurred on the upper face sheets beneath the loading rollers, and core shear failure further deepened in all CFRPHSs, as shown in Figure 12. It could be observed that the CFRPHSs with paths A, C, and D exhibited indentation failure induced by the core upper member buckling (see Figure 12a,c,d), while the indentation failure of path B was induced by the core lower member yielding (see Figure 12b). The results indicated that the antagonistic distribution and cross-distribution of paths A, C, and D endowed their nodes with good stiffness, providing better support to the loading region. Thus, the stress concentration was close to the upper face sheet in the early stage in the core members, which finally resulted in core shear failure on the upper side caused by core member buckling in the plateau stage. Their failure mode was similar to that of PPHSs. However, for path B, the symmetrical distribution and the 45° corners close to the lower face sheet had severe fiber dislocation, causing the low-stiffness nodes to be filled with pure polymer and the uneven stiffness distribution in the loading region. The uneven stiffness distribution led to a stress concentration in the core members close to the lower face sheet, and the end of core members yielded easily, ultimately resulting in core shear failure on the lower side. Thus, depending on the printing paths, the fiber distribution modes, and structural defects determined the stiffness distribution in the loading region, further affecting the stress concentration regions as well as the locations and types of core shear failure in the CFRPHSs.

In addition, some core members and face sheets touched each other with the deepening indentation failure and core shear failure, causing structural densification. Therefore, the bending loads of these CFRPHSs rose in a small amplitude after state II, as shown in Figure 8a. Moreover, combined with the microscopic morphologies after unloading in Figure 12, severe plastic deformation and damage of the polymer matrix, as well as obvious fiber breakage and fiber detachment from the matrix, could be observed in the areas of core shear failure, which caused irreversible structural failure and a decrease in the bearing capacity.

## 4. Conclusions

In this study, diamond-filled continuous fiber reinforced polymer honeycomb structures (CFRPHSs) with different printing paths were designed and fabricated via the FDM technique. The fiber dislocation at the path corners was investigated to characterize the structural defects of the CFRPHSs caused by the printing path. The bending behaviors and failure modes of the CFRPHSs with the different printing paths and pure polymer honeycomb structures (PPHSs) were studied by three-point bending tests. The results showed that the fiber distribution modes and structural defects determined the stress concentration regions and core shear failure modes of the CFRPHSs. This study revealed the mapping relationship between the printing path planning, structural defects, and structural and mechanical performance, providing references for the printing path optimization of the 3D printed CFRPHSs with high performance. The main conclusions can be drawn as follows:(1)There were three fiber distribution modes at the nodes lapped between the core and face sheets of the CFRPHSs, including symmetrical distribution, antagonistic distribution, and cross distribution, which were determined by the printing paths.(2)The structural defects at the nodes of the CFRPHSs were caused by the fiber dislocations at path corners. Low stiffness nodes filled with pure polymer caused by severe fiber dislocation led to uneven stiffness distribution in the loading region of the CFRPHSs. As the angle of the corner and the length of the straight path increased, the degree of fiber dislocation at path corners would decrease.(3)The enhancement effect of continuous fibers on the bending performance of honeycomb structures was affected by the printing paths. The path C with staggered trapezoidal distribution ensured a strong connection and good load transfer performance between the core and face sheets of the CFRPHSs, fully utilizing the supporting function of the honeycomb core, exhibiting the highest specific load capability (68.33 ± 2.25 N/g) and flexural stiffness (627.70 ± 38.78 N/mm).(4)The nodes of the CFRPHSs with fiber antagonistic distribution and cross distribution had good stiffness, providing better support to the loading region, leading to a stress concentration in the upper core and final core shear failure on the upper side caused by core member buckling. The failure mode of these CFRPHSs was similar to that of the PPHSs.(5)When the fibers were symmetrically distributed, the low-stiffness nodes filled with pure polymer caused uneven stiffness distribution in the loading region of the CFRPHSs, resulting in concentrated stress in the core members close to the lower face sheet, which ultimately led to yielding at the 45° corners.

This study has guiding significance for the printing path planning of other continuous fiber reinforced polymer two-dimensional cellular structures. It is necessary to carry out finite element simulation studies incorporating fiber path planning for further optimization of structural parameters in future work.

## Figures and Tables

**Figure 1 polymers-15-04485-f001:**
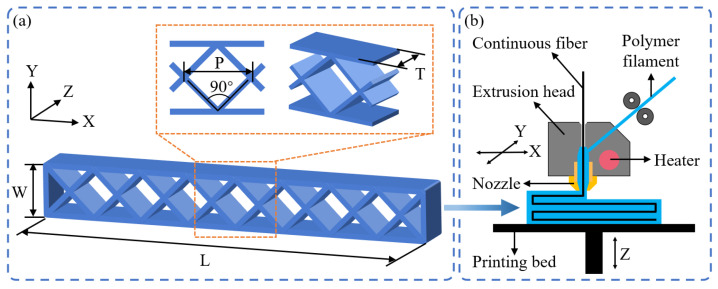
Schematic presentations of (**a**) the diamond-filled CFRPHSs with geometrical dimensions and (**b**) 3D printer using the in situ impregnation FDM technique.

**Figure 2 polymers-15-04485-f002:**
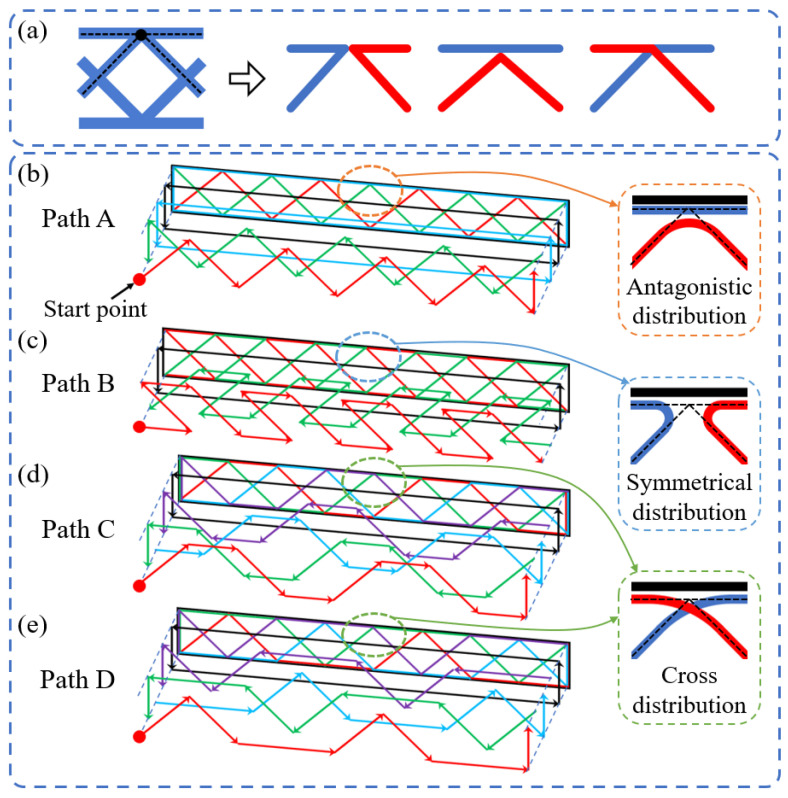
Printing path planning: (**a**) printing path strategies at the nodes, and (**b**–**e**) four printing paths of the CFRPHSs (the arrows indicate the printing directions, and the colors indicate different printing orders of continuous fibers).

**Figure 3 polymers-15-04485-f003:**
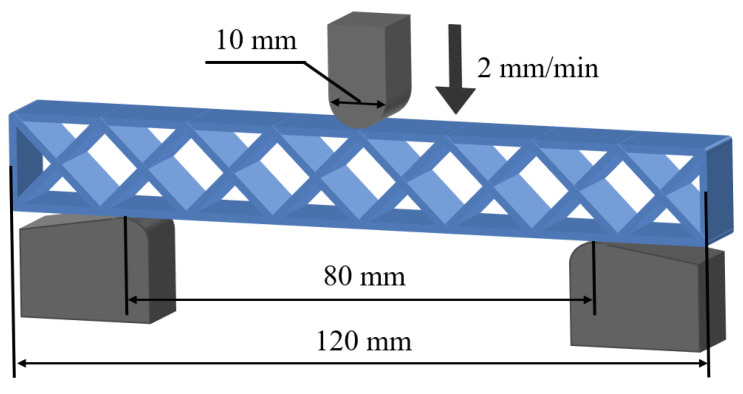
Schematic diagram of the three-point bending test (the arrow indicates the loading direction).

**Figure 4 polymers-15-04485-f004:**
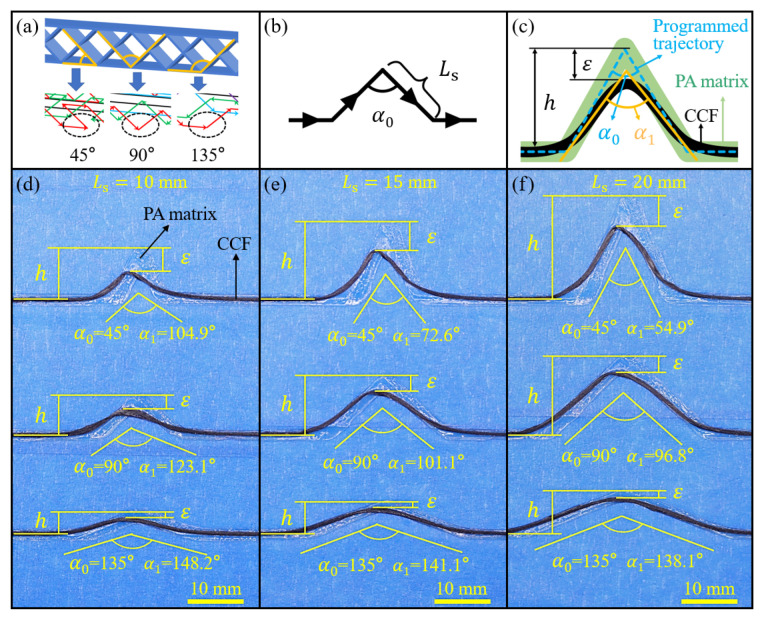
Fiber dislocation of the path corners: (**a**) three path corners in the printing paths: 45°, 90°, and 135°; (**b**) the printing paths for three path corners; (**c**) schematic diagram of the evaluation methods; geometric images of three path corners in different $L_{s}$, (**d**) $L_{s}$ = 10 mm, (**e**) $L_{s}$ = 15 mm, and (**f**) $L_{s}$ = 20 mm.

**Figure 5 polymers-15-04485-f005:**
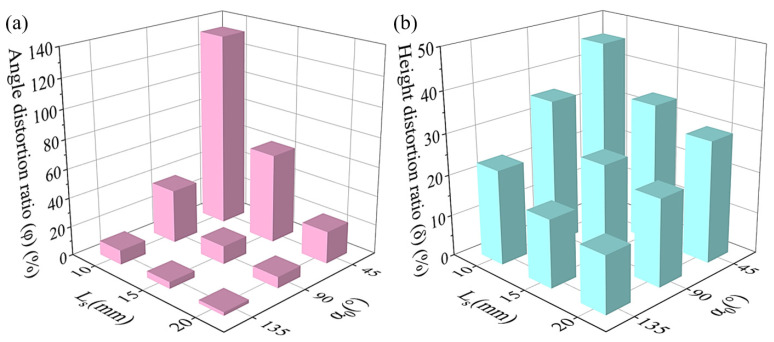
The degree of fiber dislocation at the path corners in different $L_{s}$ and $\alpha_{0}$: (**a**) angle distortion ratio (φ) and (**b**) height distortion ratio (δ).

**Figure 6 polymers-15-04485-f006:**
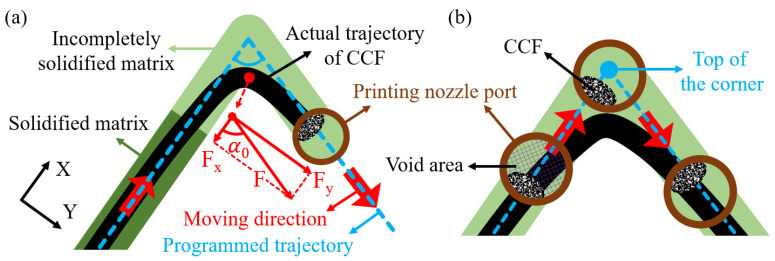
Schematic diagrams of the fiber dislocation at the path corners: (**a**) the incompletely solidified matrix could not constrain the CCF effectively; (**b**) the CCF could not maintain a fixed position relative to the printing nozzle port.

**Figure 7 polymers-15-04485-f007:**
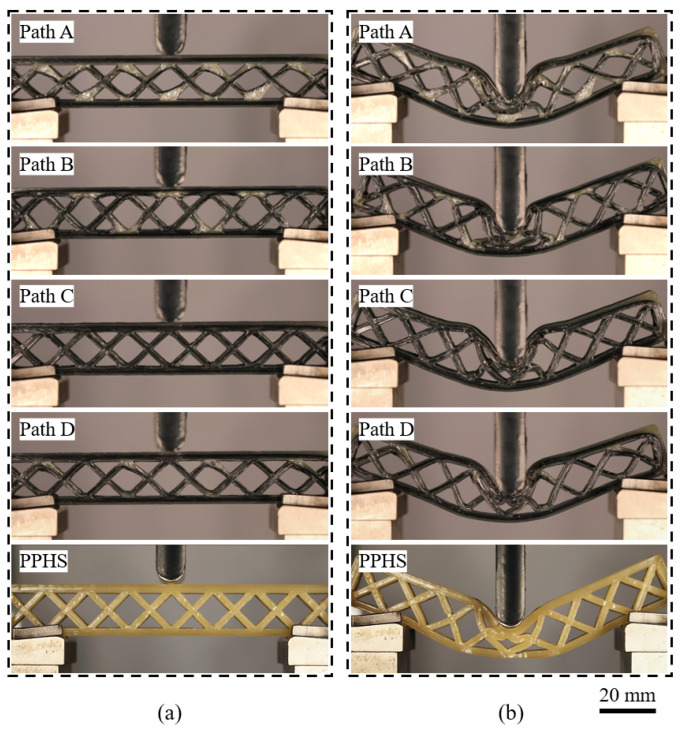
Photographs of the three-point bending test samples at displacements of (**a**) 0 mm and (**b**) 20 mm.

**Figure 8 polymers-15-04485-f008:**
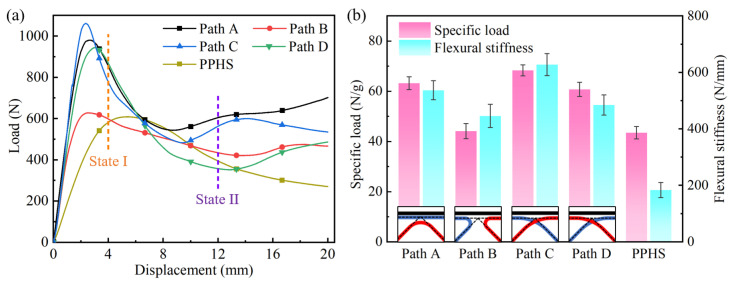
The bending properties of the CFRPHSs with different printing paths and the PPHS; (**a**) three-point bending test load-displacement curves; (**b**) specific load capability and flexural stiffness.

**Figure 9 polymers-15-04485-f009:**
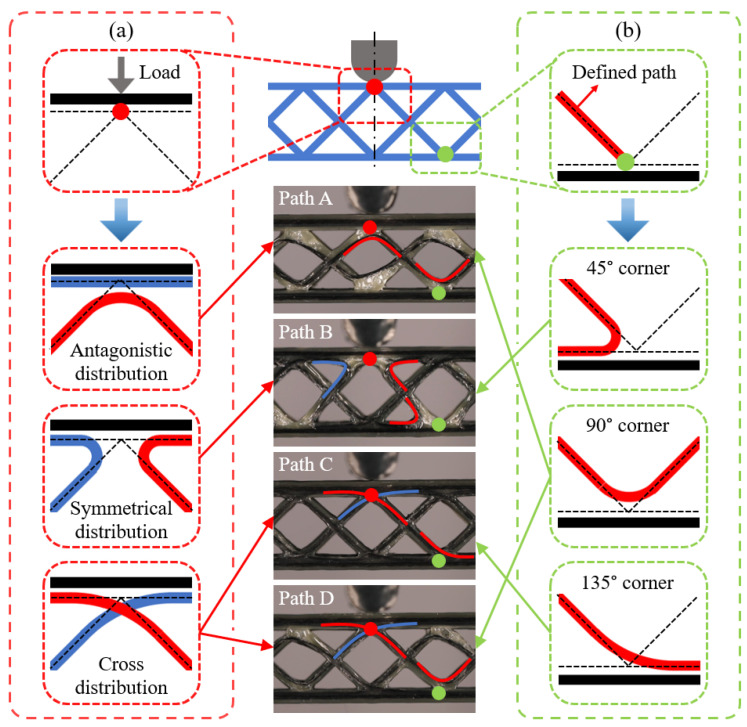
Fiber distribution modes at the nodes in the loading region of the CFRPHSs with different printing paths: (**a**) the red node beneath the loading roller; (**b**) the green node close to the lower face sheet (the red circles indicate the region beneath the loading roller, and the green circles indicate the region close to the lower face sheet).

**Figure 10 polymers-15-04485-f010:**
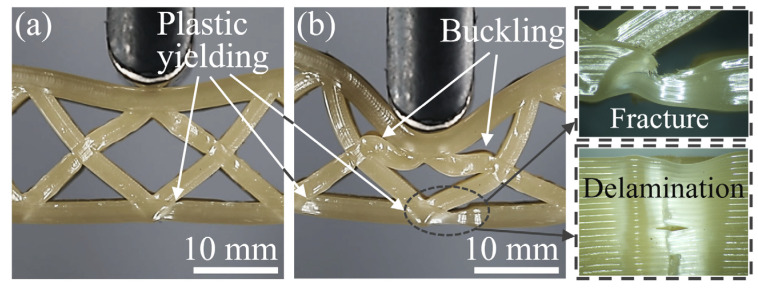
Deformation states of the PPHSs at displacements of (**a**) 6 mm and (**b**) 18 mm.

**Figure 11 polymers-15-04485-f011:**
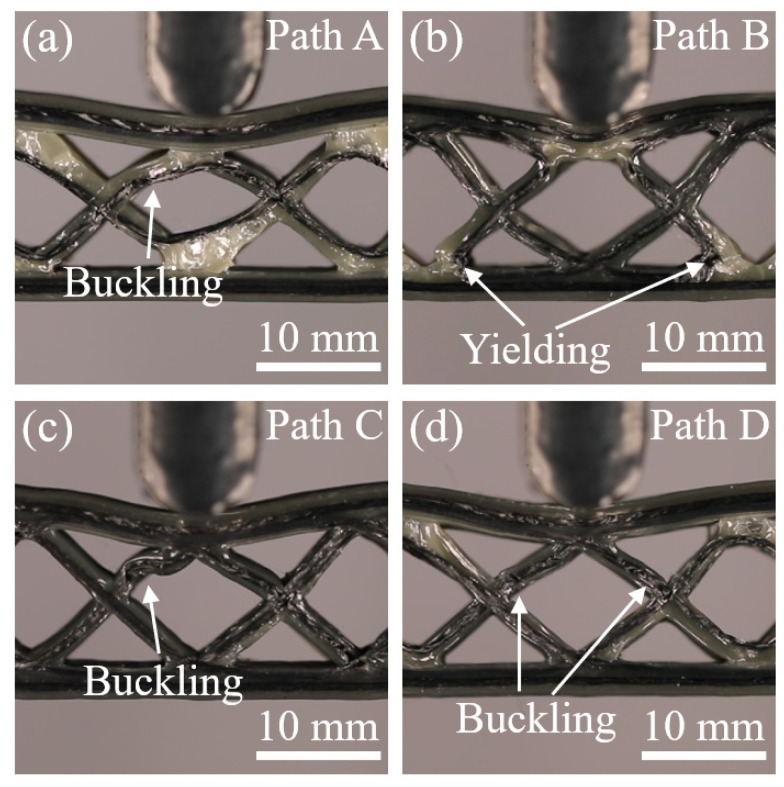
Deformation state I of the CFRPHSs with (**a**) path A, (**b**) path B, (**c**) path C, and (**d**) path D.

**Figure 12 polymers-15-04485-f012:**
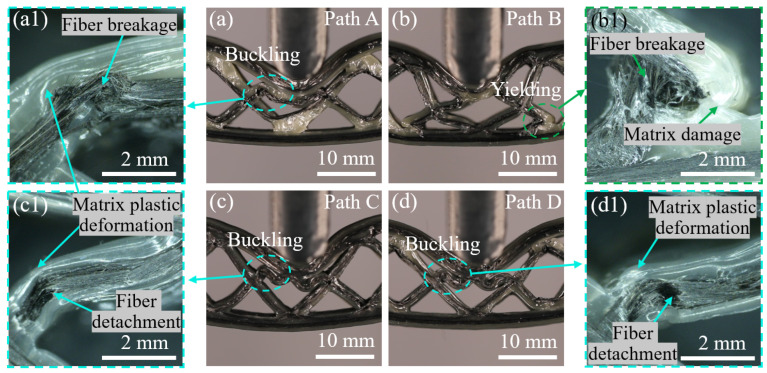
Deformation state II of the CFRPHSs with (**a**) path A, (**b**) path B, (**c**) path C, and (**d**) path D.

**Table 1 polymers-15-04485-t001:** Properties of the 3D printing PA filament and CCF.

Filament Type	Density (g/cm^3^)	Tensile Strength (MPa)	Tensile Modulus (GPa)	Elongation (%)
PA	1.12	31.40	1.05	216.50
CCF	1.77	4100.00	240.00	1.70

**Table 2 polymers-15-04485-t002:** The bending properties of all samples.

Properties	Path A	Path B	Path C	Path D	PPHSs
Specific load capability (N/g)	63.23 ± 2.53	44.12 ± 3.04	68.33 ± 2.25	60.74 ± 2.85	43.48 ± 2.46
Flexural stiffness (N/mm)	627.70 ± 33.59	696.44 ± 41.28	627.70 ± 38.78	484.56 ± 35.64	183.62 ± 26.57

## Data Availability

The data presented in this study are available on request from the corresponding author.

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
