# Peer review of "Path Planning and Bending Behaviors of 3D Printed Continuous Carbon Fiber Reinforced Polymer Honeycomb Structures"

_polymers, 2023, doi:10.3390/polym15234485_

Round 1

Reviewer 1 Report

Comments and Suggestions for Authors

The paper shows interesting results of using continuous fiber materials in AM of polyamide-based materials. Along with the manuscript there are some significant issues that need to be improved before potential publishing: 

1. Before introducing FDM technology, the authors should provide data about standardized - MEX technology (ISO/ASTM 52900).

2. What determined choosing that kind of structure - why did you choose a honeycomb instead of another available structure? You should attach a better justification. 

3. How was it possible to maintain continuously of the reinforcement along layers? Was the fiber cut after each layer? Please describe this process in a more detailed way. 

4. What determined the choice of bending test instead of compression or tensile tests? 

5. What standard have you used for bending tests? How many samples have you made for each testing series? You need to provide more data about experiment preparation. Please provide a photograph with test samples before and after bending tests. 

6. There is a statistical analysis of obtained results missing. 

7. There is a missing comparison to the matrix material (without reinforcement) which makes this paper a kind of laboratory report instead of a research paper. Also, there is a lack of proper discussion of the results. 

Reviewer 2 Report

Comments and Suggestions for Authors

In this work, an experimental study was conducted to investigate the path planning and bending behaviors of 3D-printed continuous carbon fiber-reinforced composite honeycomb structures. Some specific types of composite structures with different printing paths were investigated. A three-point bending test was used, and the data was collected and analyzed. Overall, the subject of the manuscript is of interest, but some important corrections should be made for reconsideration. The suggestions of the reviewer are as follows:

1. Some grammatical and typographical mistakes should be corrected in the manuscript.

2. In this work, the experimental study was used to investigate the 3D-printed CFRCHSs. It is noticed that the number of specimens is one of the most important parameters of the experimental study. How many specimens were used in this study?

3. There are many types of honeycomb structures. A brief review of the different kinds of honeycomb structures should be conducted to enhance the literature review.

4. A diagram of the three-point bending test should be presented in subsection 2.2.

5. The method of data collection and analysis should be clearly presented in the methods section.

6. According to Figure 6, it can be seen that the CFRCHSs with path C exhibited the highest specific load and the CFRCHSs with path B exhibited the smallest one. The physical reason should be more clearly explained. These results also suggest an optimization of the honeycomb structures; therefore, the author should discuss this point (option).

7. In this work, only an experimental study is conducted. How were the experimental results validated? In my opinion, a comparison of the experimental results and analytical results or simulation results (ANSYS or Abaqus simulation) can be added. It will make the results of the present work more reliable.

8. The conclusion should be revised; it should be consistent.

Comments on the Quality of English Language

Some grammatical and typographical mistakes should be corrected in the manuscript.

Round 2

Reviewer 1 Report

Comments and Suggestions for Authors

The authors made all sufficient corrections properly, but there is still an issue with nomenclature in the abstract part. Please fix that issue, and the paper will be ready for publication. 

Reviewer 2 Report

Comments and Suggestions for Authors

The manuscript was well revised.
